

# Modelling fatality curves of COVID-19 and the effectiveness of intervention strategies

Giovani L. Vasconcelos[1], Antônio M.S. Macêdo[2], Raydonal Ospina[3], Francisco A.G. Almeida[4], Gerson C. Duarte-Filho[4], Arthur A. Brum[2] and Inês C.L. Souza[5]

[1] Departamento de Física, Universidade Federal do Paraná, Curitiba, Paraná, Brazil
[2] Departamento de Física, Universidade Federal de Pernambuco, Recife, Pernambuco, Brazil
[3] Departamento de Estatística, Universidade Federal de Pernambuco, Recife, Pernambuco, Brazil
[4] Departamento de Física, Universidade Federal de Sergipe, São Cristovão, Sergipe, Brazil
[5] 3Hippos Data Consulting, Unaffiliated, Curitiba, Paraná, Brazil

Corresponding author
Raydonal Ospina,
raydonal@de.ufpe.br

## ABSTRACT

The main objective of the present article is twofold: first, to model the fatality curves of the COVID-19 disease, as represented by the cumulative number of deaths as a function of time; and second, to use the corresponding mathematical model to study the effectiveness of possible intervention strategies. We applied the Richards growth model (RGM) to the COVID-19 fatality curves from several countries, where we used the data from the Johns Hopkins University database up to May 8, 2020. Countries selected for analysis with the RGM were China, France, Germany, Iran, Italy, South Korea, and Spain. The RGM was shown to describe very well the fatality curves of China, which is in a late stage of the COVID-19 outbreak, as well as of the other above countries, which supposedly are in the middle or towards the end of the outbreak at the time of this writing. We also analysed the case of Brazil, which is in an initial sub-exponential growth regime, and so we used the generalised growth model which is more appropriate for such cases. An analytic formula for the efficiency of intervention strategies within the context of the RGM is derived. Our findings show that there is only a narrow window of opportunity, after the onset of the epidemic, during which effective countermeasures can be taken. We applied our intervention model to the COVID-19 fatality curve of Italy of the outbreak to illustrate the effect of several possible interventions.

## INTRODUCTION

The response interventions to the pandemic of the novel coronavirus disease (COVID-19) have varied from country to country. Several countries, especially those first hit by the disease, have adopted a standard progressive protocol, from containment to mitigation to supression (*World Health Organization (WHO), 2020*). As these strategies failed to deter the spread of the virus, government authorities introduced ever more stringent measures on their citizens' movements in an attempt to suppress or sharply reduce the propagation of the virus. More recently, countries have adopted drastic countermeasures at the very

outset of the outbreak. For example, on March 24, 2020, India announced a 3-week total ban on people 'venturing out' of their homes (*Corera, 2020*), even though there were fewer than 500 confirmed cases and only nine people had died from COVID-19 in a country with a population of 1.3 billion people. One difficulty in deciding the best approach to counter the spread of the novel coronavirus (SARS-CoV-2) is that the virus propagation dynamics is not yet well understood.

In this stark context, it becomes relevant to have simple models for the evolution of the COVID-19 epidemic, so as to be able to obtain estimates—however tentative—for the rise in the number of infected people as well as in the number of fatal cases, both in the near and in the more distant future. Such estimates are, of course, prone to high uncertainty: the less data available and the further in the future, the greater the uncertainty. Notwithstanding their inherent shortcomings, simple mathematical models provide valuable tools for quickly assessing the severity of an epidemic and help to guide the health and political authorities in defining or adjusting their national strategies to fight the disease (*Crokidakis, 2020*; *Sameni, 2020*; *Castorina, Iorio & Lanteri, 2020*; *Dehkordi et al., 2020*; *Mair et al., 2016*; *Siegenfeld & Bar-Yam, 2020*; *Bastos & Cajueiro, 2020*; *Schulz, Coimbra-AraÃojo & Costiche, 2020*; *Manchein et al., 2020*).

In the same vein, it would be desirable to have simple methods to assess the effectiveness of intervention measures as a function of the time at which they are adopted. As a general rule, the sooner an intervention is put in place the more effective it is expected to be. It is however difficult to model a priori how effective any given set of interventions will be. The effectiveness of interventions are often investigated through complex agent-based simulation models (*Ferguson et al., 2020*; *Koo et al., 2020*), which require a synthetic population and a host of parameters related to the human-to-human transmission, and as such they are very costly computationally and heavily dependent on the choice of values for the various model parameters.

In this article we use the Richards growth model (RGM) (*Richards, 1959*) to study the fatality curves, represented by the cumulative number of deaths as a function of time, of COVID-19 for the following countries: China, Brazil, France, Germany, Iran, Italy, South Korea, and Spain, which are at different stages of the epidemic. We show that the RGM describes reasonably well the fatality curves of all countries analysed in this study, except Brazil, which is in an early-to-intermediate stage of the epidemic. In the Brazilian case, we use instead the so-called generalised growth model to describe the available epidemic data. We also introduce a theoretical framework, within the context of the RGM model, to calculate the efficiency of interventions. Here an intervention strategy is modelled by assuming that its net result is captured by a change in the values of the parameters of the RGM after a given time $t_0$. In this picture, the full epidemic dynamics is described in terms of two Richards models: one before and the other after the intervention 'adoption time' $t_0$, where certain matching conditions are imposed at $t_0$. In this way, we are able to derive an analytical formula for the efficiency of the

corresponding intervention as a function of the adoption time $t_0$. We show, in particular, that the intervention efficiency decays quickly if its adoption is delayed beyond a reasonably short period of time, thus showing that time is really of essence in containing an outbreak.

## DATA

### Data source

Data used in this study were downloaded from the database made publicly available by the Johns Hopkins University (*JHU, 2020*), which lists in an automated fashion the number of confirmed cases and deaths per country. We have also compared the JHU data with the corresponding data from *Worldometer (2020)* for eventual data inconsistency and data redundancy. In all cases considered here we have used data up to May 8, 2020. In the present study we considered the mortality data of COVID-19 from the following countries: China, Brazil, France, Germany, Iran, Italy, South Korea, and Spain.

### Confirmed cases vs. mortality data

Because a large proportion of COVID-19 infections go undetected (*Li et al., 2020*), it is difficult to estimate the actual number of infected people within a given population. As many carriers of the virus are either asymptomatic or develop only mild symptoms, they will not be detected unless they are tested. In other words, the number of confirmed cases for COVID-19 is a poor proxy for the total number of infections. Furthermore, the fraction of confirmed cases relative to the total number of infections depends heavily on the testing policy of each country, which makes it problematic to compare the evolution of confirmed cases among different countries. In contrast, the number of deaths attributed to COVID-19 is a somewhat more reliable measure of the advance of the epidemic and its severity. The official numbers of deaths attributed to COVID-19 have, of course, uncertainties of their own, as countries may differ as to the criteria and protocols for recording deaths related to the disease. For instance, some countries' death figures do not include (or only later in the epidemic started to include) deaths outside hospitals, which naturally leads to under-reporting. There may also be delays in the reporting of deaths, also leading to uncertainties as to the number of deaths at a given time. Furthermore, other factors, such as the age structure of a population and quality of care, may affect the fraction of deaths relative to the number of confirmed cases. Nevertheless, taking all these factors into consideration, it is still reasonable to assume that the evolution of the number of confirmed deaths bears a relation to the dynamics of the number of infections (*Famulare, 2020*). Under these circumstances and in the absence of more reliable estimates for the number of infected cases for COVID-19, we decided here to seek an alternative approach and model the fatality curves, defined as the cumulative number of deaths as a function of time, rather than the number of confirmed cases, as is more commonly done.

## METHODS

### Mathematical models

We model the time evolution of the number of cases in the epidemic by means of the Richards growth model (RGM), defined by the following ordinary differential equation (ODE) (*Wang, Wu & Yang, 2012*; *Hsieh, 2009*):

$$\frac{dC}{dt} = rC(t)\left[1 - \left(\frac{C(t)}{K}\right)^{\alpha}\right] \tag{1}$$

where $C(t)$ is the cumulative number of cases at time $t$, $r$ is the growth rate at the early stage, $K$ is the final epidemic size, and the parameter $\alpha$ measures the asymmetry with respect to the S-shaped dynamics of the standard logistic model, which is recovered for $\alpha = 1$.

It is worth to point out that the Richards model has an intrinsic connection to the SIR epidemic model, see, for example, *Wang, Wu & Yang (2012)*. As a matter of fact, by identifying the variable $C(t)$ of the Richards model with the cumulative number of deaths of a modified SIRD model, akin to the SIR model of *Wang, Wu & Yang (2012)*, it is possible to establish a sort of 'map' between the parameters $(\alpha, r)$ of the Richards model and the parameters $(\beta, \gamma_1, \gamma_2)$ of the SIRD model, where $\beta$ is the transmission rate, $\gamma_1$ is the recovery rate and $\gamma_2$ is the death rate (*Macêdo et al., 2020*). The advantage, however, of phenomenological models, such as the Richards model, is that they allow for exact solutions (*Chowell et al., 2016*), which makes the data analysis much simpler. Furthermore, by avoiding 'the description of biological mechanisms that may be difficult to identify,' especially in an ongoing epidemic, they 'can be utilised for efficient and rapid forecasts with quantified uncertainty' (*Bürger, Chowell & Lara-Daz, 2019*). It is also worth pointing out that phenomenological growth models have been successfully applied to other epidemics, such as Zika (*Chowell et al., 2016*) and influenza (*Bürger, Chowell & Lara-Daz, 2019*), which makes these models good candidates for describing the ongoing COVID-19 epidemic, where there is still substantial uncertainty in the epidemiological parameters.

As already mentioned, in the present article we shall apply the RGM to the fatality curves of COVID-19, so that $C(t)$ will represent the cumulative numbers of deaths in a given country at time $t$, where $t$ will be counted in days from the first death. Nevertheless, in the interest of generality, in this and in the following section we shall refer to $C(t)$ simply as the number of cases.

Equation (1) must be supplemented with a boundary condition. Here it is convenient to choose

$$\ddot{C}(t_c) = 0 \tag{2}$$

for some given $t_c$, where dots denote time derivatives, that is $\ddot{C}(t) = d^2C(t)/dt^2$. A direct integration of Eq. (1) subjected to condition Eq. (2) yields the following explicit formula:

$$C(t; r, \alpha, K, t_c) = \frac{K}{\{1 + \alpha\exp[-\alpha r(t - t_c)]\}^{1/\alpha}} \tag{3}$$

where in the left-hand side we have explicitly denoted the dependence of the solution of the RGM on the four parameters, namely $r$, $\alpha$, $K$, and $t_c$. In fitting Eq. (3) to empirical data it is convenient to set $C(0) = C_0$, where $C_0$ is the number of deaths recorded at the first day that a death was reported. Using that $C(0) = K/[1 + \alpha \exp(\alpha r t_c)]^{1/\alpha}$, we can eliminate $t_c$ in favour of the other parameters, so that we are left with only three free-parameters, namely $(r, \alpha, K)$, to be numerically determined.

We note, however, that the RGM is not reliable in situations where the epidemic is in such an early stage that the available data is well below the estimated inflection point $t_c$, that is, when the epidemic is still in the so-called exponential growth regime (*Wu et al., 2020*). In this case, it is preferable to use the so-called generalised growth model (*Wu et al., 2020*; *Chowell, 2017*), which is defined by the following ODE:

$$\frac{dC}{dt} = r[C(t)]^q \tag{4}$$

where the parameter $q$ provides an interpolation between the sub-exponential regime $(0 < q < 1)$ and the exponential one $(q = 1)$. The solution of Eq. (4) is

$$C(t; r, q, A) = [A + (1 - q)rt]^{1/(1-q)} = A^{1/(1-q)} e_q(rt/A) \tag{5}$$

where the function $e_q(x) = [1 + (1 - q)x]^{1/(1 - q)}$ is known in the physics literature as the $q$-exponential function (*Tsallis, 1988*; *Picoli et al., 2009*). Here the parameter $A$ is related to the initial condition, that is $A = C(0)^{(1 - q)}$, but we shall treat $A$ as a free parameter to be determined from the fit of Eq. (5) to a given dataset.

## Intervention strategy and efficiency

We define an intervention strategy in the context of the RGM by considering that the corresponding measures, as applied to the actual population, induce at some time $t_0$ a change in the parameters of the model. In this way, the solution for the whole epidemic curve acquires the following piecewise form:

$$C(t) = \begin{cases} C(t; r, K, \alpha, t_c), & t \leq t_0 \\ C(t; r', K', \alpha', t_c'), & t > t_0 \end{cases} \tag{6}$$

where we obviously require that $K'/K < 1$. Furthermore we impose the following boundary conditions at $t_0$:

$$C(t_0; r, K, \alpha, t_c) = C(t_0; r', K', \alpha', t_c') \tag{7}$$

$$\dot{C}(t_0; r, K, \alpha, t_c) = \dot{C}(t_0; r', K', \alpha', t_c') \tag{8}$$

Note that condition Eq. (8) takes into account, albeit indirectly, the fact that intervention measures take some time to affect the epidemic dynamics. In other words, the trend (i.e. the derivative) one sees at a given time $t$ reflects in part the measures taken at some earlier time (or lack thereof). Thus, imposing continuity of the derivative of the epidemic curve at time $t_0$ in our 'intervention strategy' seeks to capture this delay effect.

In our intervention model above, we assume that the net result of the intervention is to alter the parameters $r$ and $\alpha$ of the RGM after the time $t_0$; see Eq. (6). In other words,

we assume that the parameters of the RGM can capture (albeit in an effective and simplified manner) some basic aspects of the underlying epidemic dynamics, so that changes in the mechanisms of the disease propagation—owing, say, to the introduction of intervention measures—could be described in terms of variations in the model parameters. It is in this sense that we associate actual interventions with possible changes in the model parameters, assuming of course that the growth model is still valid after the interventions. Admittedly, the difficult part is to estimate how a particular set of intervention measures (e.g. social distancing, contact tracing and quarantine, school closures, etc.) would influence these parameters. This link between actual interventions and the RGM parameters cannot, of course, be obtained within the context of the Richards model alone. It requires, for example the use more complex models, such as agent-based or compartmental models, to implement specific interventions, after which one can use the RGM to fit the resulting epidemic curves; see, for example *Chowell (2017)* where a similar approach was adopted albeit in the context of quantifying the uncertainty in epidemiological parameter estimates. Comparison of the RGM parameters after the intervention with those for the reference curve (i.e. without intervention) could thus shed light on how actual interventions can be mimic within the RGM approach. We are currently pursuing this strategy—namely, using the RGM formula to fit simulations from both agent-based and SIR-type models—to gain a better understanding of how interventions can be reflected in the parameters of the RGM. Such an analysis, however, is still ongoing (*Macêdo et al., 2020*) and is beyond the scope of the present article.

Note that, as discussed above, the time $t_0$ is not the actual time of adoption of the intervention but rather the time after which the corresponding measures start to affect the epidemic dynamics, as reflected in a change in the evolution of the number of cases. Nevertheless we shall for simplicity refer to $t_0$ as the intervention 'adoption time.' We remark, furthermore, that as far as interventions go, there are basically two aims: (i) to reduce the speed of an epidemic, which is essentially to 'flatten the curve' of daily deaths and (ii) to reduce the epidemic size, that is the total number of deaths due to the epidemic. In this article we address the latter case, since the interventions modelled by the strategy Eq. (6) are designed to yield $K' < K$. In other words, the aim here is to 'bend' the cumulative curve of deaths as soon as possible, so as to reach a lower plateau at the end of the epidemic (for more details, see "Discussion").

As defined in Eq. (6), an intervention strategy adopted at time $t_0$ can be viewed as a map $(r, K, \alpha, t_c) \rightarrow (r', K', \alpha', t_c')$ in the parameter space of the RGM, which results in the condition $K'/K < 1$. Let us therefore define the intervention efficiency as the relative reduction (expressed in percentage) of the total number of cases: $\eta(t_0) = (K - K')/K$, where it is assumed that $\eta(t_0) > 0$. Using conditions Eqs. (7) and (8) in Eq. (3), one obtains that

$$\eta(t_0) = 1 - \frac{y}{[1 - x(1 - y^\alpha)]^{1/\alpha'}} \tag{9}$$

where $y = C(t_0)/K$ and $x = r/r'$. Later we will exemplify the above measure of intervention efficiency, using as input the parameters $r$ and $\alpha$ obtained from the fatality curve of the COVID-19 from Italy. This will allow us to investigate how the efficiency of different strategies (i.e. for different choices of $r'$ and $\alpha'$) varies as a function of the adoption time $t_0$.

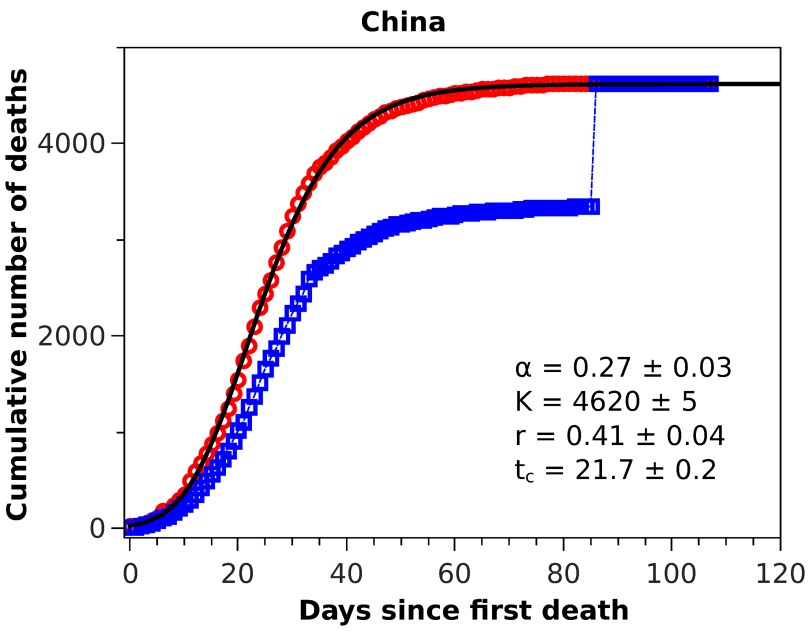

**Figure 1 Cumulative number of deaths attributed to COVID-19 for China up to May 8, 2020.** The blue squares represent the official numbers, where one sees a 'discontinuity' at $t = 84$, owing to the revision of the data announced by China on April 17, 2020. The red circles are a renormalisation of the official data prior to the revision, according to the procedures indicated in the text. The solid black curve is the fit to the renormalised data by the Richards growth model.

## Statistical fits

All statistical fits in the article were performed using the Levenberg–Marquardt algorithm (*Moré, 1978*) to solve the corresponding non-linear least square optimisation problem. In the case of the Richards model, we set $C(0) = C_0$, where $C_0$ is the number of deaths recorded at the first day that a death was reported, so that there remain three parameters, namely $(r, K, \alpha)$, to be determined. For the $q$-exponential growth model we also need to determine three parameters $(r, q, A)$. The fitting procedures were implemented in the opensource software QtiPlot, which was also used to produce the corresponding plots in Figs. 1–3. The plots in Fig. 4 were produced with the data visualisation library Matplotlib for Python.

# RESULTS

## Fatality curves

In Fig. 1 we show the official cumulative number of deaths (blue symbols) attributed to COVID-19 for China, where it is clearly visible the jump on day 84 from the first death, when the death toll was revised upward by almost 50%. Since officials in Wuhan, China, informed only that this increase 'reflected updated reporting and deaths outside hospitals' (*BBC News, 2020*), it does not seem possible to reconstruct the actual fatality curve for China; nor does it make much sense to fit any model to the data prior to this correction, owing to their unreliability. It is possible nonetheless to render the Chinese data amenable to a statistical fit, if only as a test of the model, if one somehow smooths the
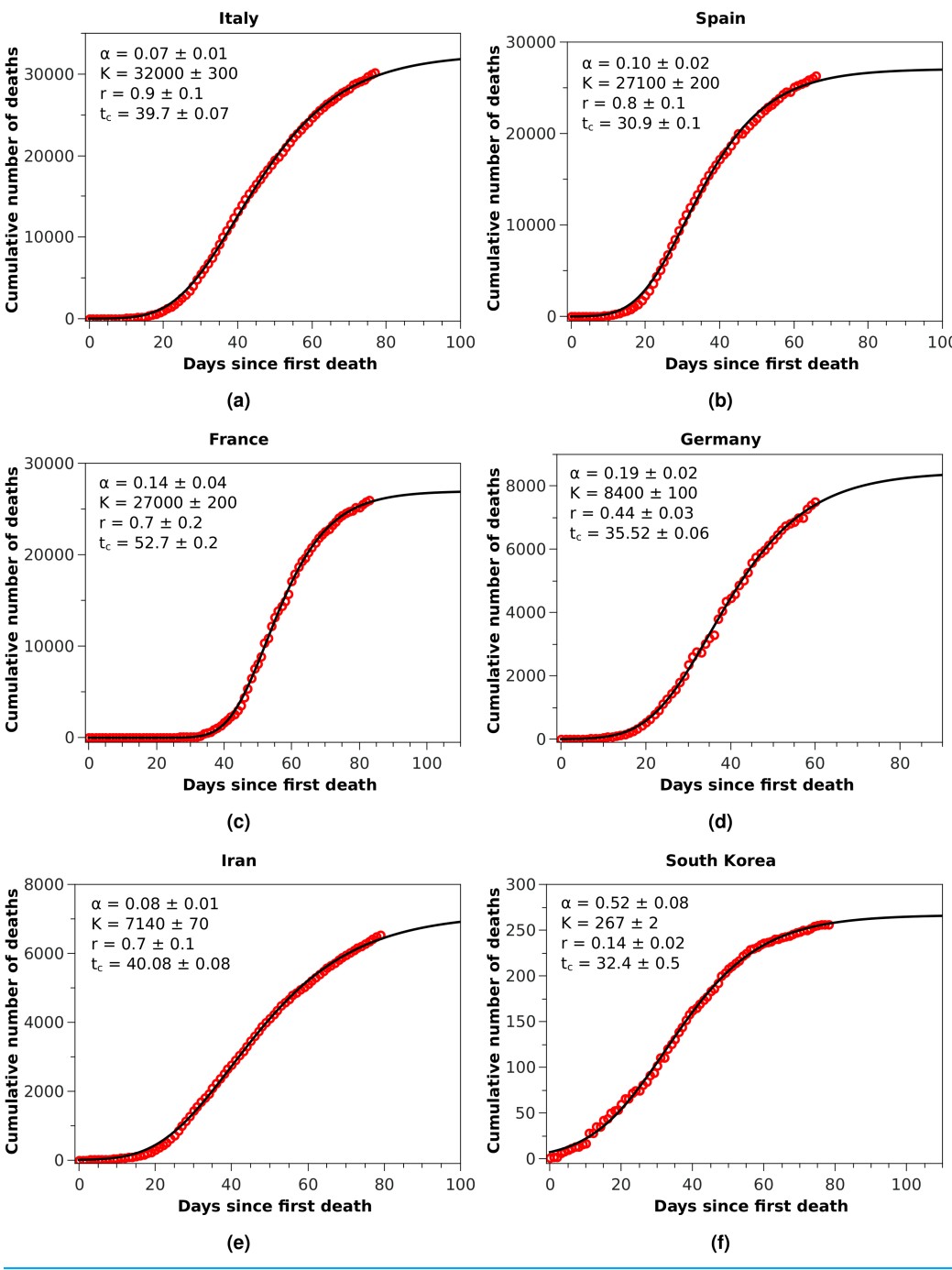

**Figure 2** Cumulative number of deaths (red circles) attributed to COVID-19 up to May 8, 2020, for (A) Italy, (B) Spain, and (C) France, (D) Germany, (E) Iran, and (F) South Korea. The solid black curves are the fits by the Richards growth model, see Eq. (3), with the corresponding parameters given in the respective insets.                                                 

revised data. In this spirit, we adopted the following ad hoc procedure to obtain a smooth 'empirical' curve for China: we multiplied all data points prior to the revision date by a factor corresponding to the ratio between the numbers of deaths after and before the revision date, meaning that the excess deaths due to the data correction was uniformly

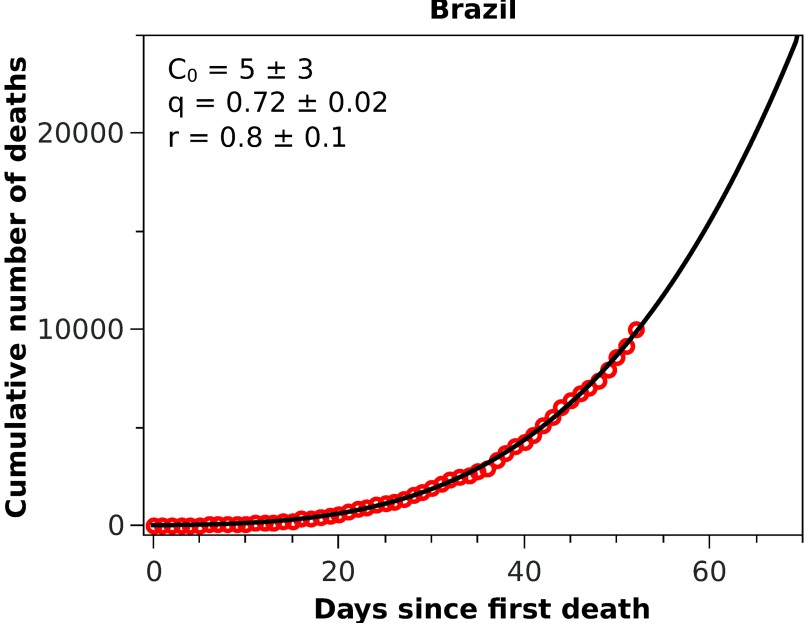

**Figure 3 Cumulative number of deaths attributed to COVID-19 up to May 8, 2020, for Brazil.** The solid black curve represents the fit with the $q$-exponential model shown in Eq. (5), with the parameters given in the inset.

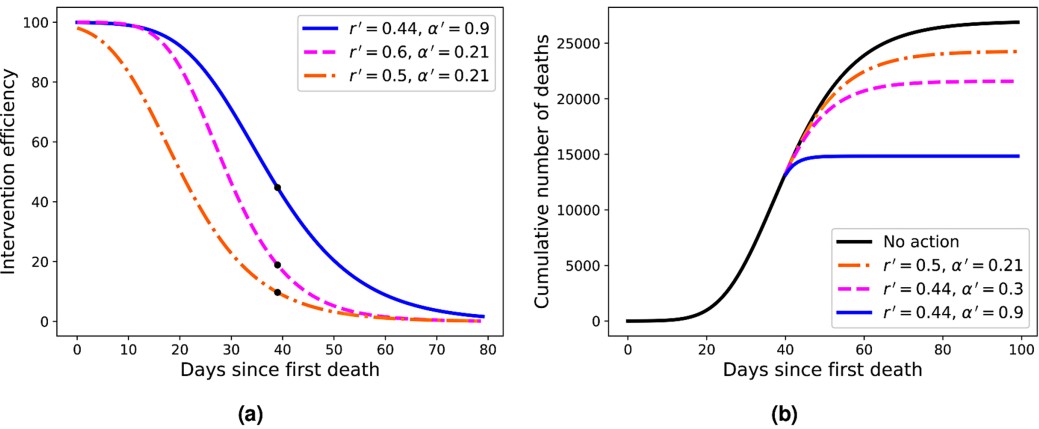

(a)                                                          (b)

**Figure 4 (A) Efficiency curves as a function of the adoption time $t_0$ for two different intervention strategies applied to Italy's fatality curve up to April 1, 2020, for which we obtained $r = 0.44$ and $\alpha = 0.21$.** Here the parameters are (from top to bottom) $r' = 0.44$ and $\alpha' = 0.9$ (blue); $r' = 0.6$ and $\alpha = 0.21$ (green); and $r' = 0.5$ and $\alpha' = 0.21$ (red). (B) Fatality curves corresponding to the three intervention indicated by the black dots in (A), as compared to the no-action reference curve (black uppermost curve).

redistributed by the same proportion for all dates before the revision. This is admittedly an arbitrary procedure (perhaps a worst case scenario in terms of change of shape of the unknown true curve), which is intended only as a numerical way of 'welding' the two sides of the curve at the jump. This 'renormalised' fatality curve (red circles) for China is shown in Fig. 1, superimposed with the corresponding statistical fit of the RGM curve (black solid line), where the fit parameters are shown in the inset of the figure. One sees from the

figure that the fatality curve in this case, where the epidemic has apparently levelled off, is well adjusted by the RGM formula.

The good performance of the RGM for the renormalised data from China, exhibited in Fig. 1, encouraged us to apply the model to other countries at earlier stages of the epidemic. Here, however, care must be taken when estimating the model parameters from small time series, since it is well known that the Richards model (*Wang, Wu & Yang, 2012*; *Hsieh, 2009*) and its variants (*Wu et al., 2020*) are susceptible to the problem of *over fitting*, owing to the redundancy of the parameters. This may lead, for example, to estimation of certain parameters that are outside of biologically or otherwise reasonable ranges. For example, when applied to the number of infected cases in an epidemic, the parameter $\alpha$ should be constrained within the interval (0,1) (*Wang, Wu & Yang, 2012*). Here we apply the RGM instead to the number of deaths, but we assume that the same constraint should be observed. In other words, fits that return $\alpha$ outside this interval are disregarded as not reliable. Similarly, we restrict the acceptable values of $r$ to the range (0,1), as we observed that values of $r$ outside this interval tends to be an indication that the RGM is not quite suitable to fit the data. In other words, for our purposes here we assume that the restrictions $0 < r < 1$ and $0 < \alpha < 1$ are useful empirical criteria for the validity of the RGM, which can also be verified from the map between the Richards model and the modified SIRD model (*Macêdo et al., 2020*). The unsuitability of the RGM is particularly evident when the available data does not encompass the inflection point $t_c$ (*Chowell, 2017*; *Wu et al., 2020*), although as more data is accumulated the model is expected to become more accurate. As an empirical criteria, we thus consider here that the RGM is only acceptable if $t_c$ is smaller than the time of the last data point; if this criteria is not fulfilled by a particular dataset, we then apply the generalised growth model as given in Eq. (5).

With these considerations in mind, we applied the RGM to the mortality data of COVID-19 from several other countries. In Fig. 2, we show the cumulative deaths for Italy, Spain, France, Germany, Iran, and South Korea, together with the respective RGM fits. Here again the RGM seems to be able to provide a reasonably good fit to the data for all the above countries. In the case of Brazil, where the epidemic curve has not yet reached its inflection point, the RGM is not justifiable. In such cases, it is more advisable to use a simpler growth model, such as the $q$-exponential. In Fig. 3 we show the fit of the $q$-exponential curve Eq. (5) to the Brazilian data, where one sees that the agreement is very good. From the fit we get $q = 0.72$, indicating that the fatality curve is in a sub-exponential growth regime.

### Intervention efficiency

As already mentioned, an intervention strategy in our model is defined by the two parameters $r'$ and $\alpha'$ of the new Richards model after the adoption time $t_0$; see Eq. (6). It is premature at this stage to establish a more direct link between actual intervention measures and a corresponding change in the parameters of effective growth models, as already discussed. As a matter of fact, we are currently pursuing this connection between our phenomenological model of interventions and concrete measures (such as social distancing, quarantine, school closures, etc.) by implementing these measures in more

complex models, such as agent-based population models and SIRD-type models, and then use the RGM to fit the resulting fatality curves with and without interventions. In this way, we hope to establish how a possible combination of variations in the parameters $(r, \alpha)$ can mimic (at least approximately) a given real intervention. Such an endeavour, however, is beyond the scope of the present article. Our main goal here is to introduce a quantitative measure of the effectiveness of interventions (in the context of the Richard model) and highlight some of its main and important features. More specifically, we shall take a reference curve from the RGM, that is, with a given set of parameters $(r, \alpha, t_c, K)$, consider different intervention 'strategies' as defined in Eq. (6), and then discuss their respective efficiencies.

As the reference RGM curve we take here that obtained by fitting the Italian data up to April 1, 2020, when Italy was somewhat in the middle of the outbreak (this corresponds to $t = 40$ in Fig. 2A), which gave $r = 0.44$ and $\alpha = 0.21$. In Fig. 4A we show the efficiency curves as a function of the adoption time $t_0$ for three different interventions, namely: (i) $r' = 0.5$ and $\alpha' = 0.21$ (red dot-dashed curve); (ii) $r' = 0.6$ and $\alpha = 0.21$ (green dashed curve); and (iii) $r' = 0.44$ and $\alpha' = 0.9$ (blue solid curve). In Fig. 4B we show the resulting fatality curves, as compared to the no-action case (black curve), after implementing the three intervention actions indicated by the black dots on the red, green, and blue curves of Fig. 4A.

## DISCUSSION

We have seen above that the RGM describes rather well the fatality curves of COVID-19 from different countries, which are at different stages of the pandemic. For example, in the case of China, whose fatality curve has pretty much levelled off indicating a near-end of the epidemic, the Richards Eq. (3) fits rather well the entire epidemic curve. Here, however, because of the upward revision of the Chinese mortality data on April 17, 2020, we had to adopt an ad hoc data-correction procedure, as explained above, so as to smooth the 'discontinuous' empirical curve and thus render it amenable to a mathematical description. In spite of the fact that we were therefore forced to work with rescaled data for the case of China, the good fit provided by the RGM observed in Fig. 1 may be seen as a good indication of the validity of the RGM to describe mortality data of COVID-19 for the full epidemic course. The RGM was also in good agreement with the empirical data for other countries, such as Italy, Spain, France, Germany, Iran, and South Korea. In all these cases, the last data point is sufficiently beyond the inflection point $t_c$ obtained from the fit of the RGM Eq. (3) to lend some credibility to the model predictions.

As emphasised earlier, we considered here that a statistical fit with the RGM is only acceptable if $t_c$ is smaller than the time of the last data point. In addition, we used two additional admissibility criteria, namely that the parameters $r$ and $\alpha$ should be both in the interval $(0,1)$. For countries where the epidemic is still in a relatively early stage, these criteria are usually not met. For example, when we applied the RGM to the mortality data from Brazil, we found that all three empirical criteria above were not satisfied, indicating that the fit with the RGM was 'premature,' as there were not yet enough data point in the Brazilian fatality curve to make reliable estimations of the model parameters. In this

situation, we then fitted the Brazilian data with the $q$-exponential function given in Eq. (5) and found $q < 1$, indicating that the fatality curve is in a sub-exponential growth regime. This slower-than-exponential growth probably stems from the mitigation actions that have been put in place in Brazil since the onset of the outbreak; similar effect (i.e. sub-exponential growth) was also observed, for example, in China, although for the number of cases, and it was attributed to containment policies as well (*Maier & Brockmann, 2020*). From the fit parameters, see inset of Fig. 3, we predict that the current time for doubling the number of deaths in Brazil is 13 days, which is considerably higher than the value of about 5 days we obtained in the first version of this study (which used data up to April 1, 2020). However, the fact that there is no clear indication that the Brazilian fatality curve is near the inflection point is a cause of concern.

As discussed above, the RGM fits rather well the mortality data of several countries for which the epidemic is still ongoing. We emphasise, however, that our primary interest in the RGM is not so much aimed at its predictive capacity in the face of incomplete data, but rather more so as a mathematical framework in which one can obtain quantitative measures (in fact, an explicit formula) for the effectiveness of mitigation strategies. Using this framework, we were able to compute an explicit formula for the efficiency of an intervention as a function of the adoption time $t_0$, some example of which are shown in Fig. 4A. From these illustrative examples, several important consequences of our efficiency formula can be obtained—which we believe are of general applicability, at least in a qualitative sense.

For example, one sees from Fig. 4A that in order to attain an efficiency of at least 80% the three interventions shown in the figure must be adopted up to the times $t_0 = 11, 21$, and 26, respectively. However, if the interventions are further delayed by ten more days the respective efficiencies drop to about 50% or less, in all cases. Furthermore, a delay of additional 20 days above the time-window of 80% efficiency brings the efficiency to less than 30% in all cases exemplified in Fig. 4A. This shows that, in general, delaying interventions beyond a reasonable early period of time—the so-called window of opportunity—can have the adverse effect of reducing considerably the effectiveness of an intervention. As for the role of the parameters $r$ and $\alpha$ in affecting the intervention efficiency, one more careful analysis of the model shows that a larger $r$ (with $\alpha$ kept fixed) implies a smaller inflection time $t_c$, which in turn leads to a smaller $K'$, and hence greater efficiency. A similar but stronger effect is obtained with increasing $\alpha$ (for $r$ fixed): the larger the $\alpha$, the sooner the curve 'bends' toward the plateau, thus yielding a lower final death toll. In fact, the control in the asymmetry of the 'S-shaped' curve, and thus the value of $t_c$, was the original motivation for introducing the parameter $\alpha$ in the Richards model. It is thus natural to expect a stronger effect in changing $\alpha$ (for $r$ fixed) than changing $r$ (for $\alpha$ fixed). It is also clear from Fig. 4 that stronger interventions (e.g. with higher values of $\alpha$) provide wider windows of opportunity, which makes epidemiological sense.

For the three examples of interventions shown in Fig. 4A, we illustrated in Fig. 4B the fatality curves resulting from theses interventions, when they are adopted at the particular time indicated by the black dots on the curves of Fig. 4A. It is interesting to observe that the qualitative behaviour seen in Fig. 4B for the fatality curves after the interventions is

reminiscent of similar curves, albeit for the number of infections, obtained in agent-based simulations of actual intervention measures (*Weng & Ni, 2015*). This may be seen as further evidence that the Richards model can indeed be extended to model interventions, as proposed above.

## CONCLUSIONS

To summarise, this article provides important insights into the time evolution of the accumulated number of deaths attributed to COVID-19, especially for countries that are in the middle of the outbreak or only recently have passed the inflection point in the curve of accumulated deaths. Our modelling of the fatality curves is particularly relevant for the COVID-19 epidemic because the actual number of infections in this case remains largely unknown, and so one is forced to deal with proxy measures, such as mortality data, to gain a better understanding of the actual severity of the epidemic.

The article also shows how simple and soluble mathematical models can provide a rich theoretical framework in which to investigate some basic and deep aspects of epidemic dynamics. In particular, we have successfully applied the Richards growth model to describe the fatalities curves of seven different countries at different stages of the COVID-19 outbreak, namely China, France, Germany, Italy, Iran, South Korea, and Spain. We also analysed the case of Brazil, which is in an earlier stage of the outbreak, and so we had to resort to a modified exponential growth model, known as the generalised growth model. This model also gave a good fit of the rising fatality curve of Brazil, from which we could estimate that the current time for doubling the number of fatalities from COVID-19 is 13 days.

Another important contribution of the present study is to provide an analytical formula to quantitatively assess the efficiency of intervention measures in an ongoing epidemic. Interventions strategies were defined in the context of the Richards model as a change in the model parameters at some specified time, referred to as the intervention adoption time. Our formula shows that, in general, the efficiency of an intervention strategy decays quite quickly as the adoption time is delayed, thus showing that time is really of essence in containing an outbreak. The present work can be extended in several ways. For example, a direct connection with a SIRD-type model can be explored in order to find the underlying epidemiological meaning of the parameters $r$ and $\alpha$ of the RGM. Another possible direction of research consists in seeking to identify how the parameters of the RGM can be varied, perhaps even continuously in time, so as to mimic the effect of actual intervention measures. One could then apply such an extended Richards model with time-dependent parameters to improve its fitting performance whenever needed. All these interesting research avenues are currently being explored in our group.

### Funding

This work was supported by Conselho Nacional de Desenvolvimento Científico e Tecnológico (CNPq) under Grants No. 303772/2017-4 (GLV), No. 312612/2019-2

(AMSM), No. 305305/2019-0 (RO) and Coordenação de Aperfeiçoamento de Pessoal de Nível Superior (CAPES) of Brazil. The funders had no role in study design, data collection and analysis, decision to publish, or preparation of the manuscript.

## Grant Disclosures

The following grant information was disclosed by the authors:
Conselho Nacional de Desenvolvimento Científico e Tecnológico (CNPq): 303772/2017-4, 312612/2019-2 and 305305/2019-0.
Coordenação de Aperfeiçoamento de Pessoal de Nível Superior (CAPES).

## Competing Interests

The authors declare that they have no competing interests. Inês Cristina Lemos de Souza is the founder and director of 3Hippos Data Consulting (https://www.3hippos.com.br) in Brazil, which specialises in providing data analysis and training in data literacy for not-for-profit social organisations. She has no affiliations with or involvement in any organisation or entity with any financial interest, or non-financial interest in the subject matter discussed in this article. Giovani Lopes Vasconcelos and Inês Cristina Lemos de Souza are a married couple and have worked together with the rest of the team in a collaborative fashion.

## Author Contributions

- Giovani L. Vasconcelos conceived and designed the experiments, performed the experiments, analysed the data, prepared figures and/or tables, authored or reviewed drafts of the paper, and approved the final draft.
- Antônio M.S. Macêdo conceived and designed the experiments, performed the experiments, analysed the data, prepared figures and/or tables, authored or reviewed drafts of the paper, and approved the final draft.
- Raydonal Ospina conceived and designed the experiments, performed the experiments, analysed the data, prepared figures and/or tables, authored or reviewed drafts of the paper, and approved the final draft.
- Francisco A.G. Almeida conceived and designed the experiments, performed the experiments, analysed the data, prepared figures and/or tables, authored or reviewed drafts of the paper, and approved the final draft.
- Gerson C. Duarte-Filho conceived and designed the experiments, performed the experiments, analysed the data, prepared figures and/or tables, authored or reviewed drafts of the paper, and approved the final draft.
- Arthur A. Brum conceived and designed the experiments, performed the experiments, analysed the data, prepared figures and/or tables, authored or reviewed drafts of the paper, and approved the final draft.
- Inês C.L. Souza conceived and designed the experiments, performed the experiments, analysed the data, prepared figures and/or tables, authored or reviewed drafts of the paper, and approved the final draft.
## Data Availability

The codes in Python and the dataset are available in the Supplemental Files.

## Supplemental Information

Supplemental information for this article can be found online at http://dx.doi.org/10.7717/peerj.9421#supplemental-information.

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
