# Peer review of "Modelling fatality curves of COVID-19 and the effectiveness of intervention strategies"

_PeerJ, doi:10.7717/peerj.9421_

## Round 0.1 · original submission · Major Revisions

Your manuscript elicited a wide range of opinions about its validity, especially about the appropriateness and the simplicity of the models you selected. Although you will need to respond to all the author comments in detail, here are ones that I want to particularly highlight:

1. You need to better explain how interventions affect models, and justify your choices of r and alpha. This means tying in the mathematical components of your models with the actual epidemiologic importance of them.

2. Add other countries (South Korea, Singapore), and address where model does not do an effective job. One reviewer indicated that your models do not work well for South Korea. If this is true, then this should be explored and explained, as it is important to know under what circumstances (and countries) the models do and do not predict mortality well.

3. One reviewer made this important comment: "Their mathematical models (the Richards growth model (RGM) and generalized growth model) are too simple to capture country-specific COVID-19 transmission dynamics and also country-specific fatalities in the five countries." Please provide a strong justification for your models in addressing this comment, because interventions will be country-specific, so models should be specific to countries.

4. Another reviewer wrote: "This is a basic exercise in curve fitting and does not provide any insight to the nature of the epidemic. Since it is a phenomenological model, it is very strange to interpret a change in parameters as interventions. Interventions should modify the mechanisms of disease spread in certain ways, but such mechanisms are not considered here. Even worse, in reality there were a sequence of progressively applied interventions in each of the considered countries." Again, your models should be more than curve-fitting exercises, so it will be important to integrate the epidemiology into the statistical models. Otherwise, your models will be exercises into historic curve fitting, but will not help predict how interventions can affect future mortality in countries.

Reviewer 1 ·

Basic reporting

The paper is well written and presented.

Experimental design

The authors fit Richard's model to cumulative death data of COVID-19. This is a rather flexible family of curves that can be fitted to practically any sigmoid curve, and cumulative epidemic data often (but not always) have a sigmoid shape.
This is a basic exercise in curve fitting and does not provide any insight to the nature of the epidemic. Since it is a phenomenological model, it is very strange to interpret a change in parameters as interventions. Interventions should modify the mechanisms of disease spread in certain ways, but such mechanisms are not considered here. Even worse, in reality there were a sequence of progressively applied interventions in each of the considered countries.

Validity of the findings

There are no real scientific findings in this article. The Richard's model spectaculartly fails on Korean data (which was not presented here).

Reviewer 2 ·

Basic reporting

I find the basic reporting of the study to be very descriptive and realistic. It made
me feel that I had a good understanding of what was going to be examined. The language of writing is easy to understand even to the audience outside computational/mathematical modeling field.

Experimental design

It's interesting to see the modeling the COVID-19 development of several countries in different stages, including China, Italy and Iran, which brings a nice selection of target subjects.

I believe it would be also reasonable if you could include more countries like Singapore and/or South Korea, especially in the "intervention" part of the experiment, as these countries have taken relevant approaches to control the virus situation

Validity of the findings

I believe that the findings are valid, but the further significance needs to be considered. Is there existing experiments with similar approaches, in dealing with COVID-19 or even non-COVID-19 respiratory infectious disease like influenza? From my point of view, an extra content comparing the current research with other relevant modelings of respiratory infectious disease would be of great help to shed light to the community fighting against the COVID-19.

Additional comments

The current manuscript described the of establishment of a mathematical/computational model of the COVID-19 in aspects of fatality and how it respond to interventional strategies. I believe it's a well written article and it follows nicely to the scope of PeerJ journal publication. The topic is interesting and it could help improve the understanding of the disease development and provide sights to prevention and responding measures. I believe the the submission would be of significance to the community fighting against COVID-19, and how helps draw the attention of disease/disaster response community to the computer science and simulation world.

·

Basic reporting

The paper follows a format close to the Standard structure, there is a background section but no Introduction replaced with a Background section and the materials section replaced with a section data. It might be worth reverting to having an introduction rather than background section to emphasise the point that this paper is about assessing interventions rather than predicting the course of specific outbreaks and that those fits are designed to validate the use of the RGM. In general, however, the structure is logical and clear, the language is approachable and appropriated, there are sufficient references.
The figures are reasonable, though the use of blue and red will make it difficult for those who need to print and view the papers in black and white. Using colours which have different intensities or different line types or widths, or symbols might help with this.
The raw data has been supplied and the code is correct.


Line 85 – confirmed deaths are more reliable, but “much more reliable” is a strong statement and would need some justification, particularly given that the discussion suggests that timing is critical and there are known to be variable delays in the reporting of deaths

Experimental design

Whilst based on a mathematical model, the focus of the manuscript is on the use of that model to investigate interventions into a pandemic thus it fits well with the scope of the journal. The subject is topical and the research timely. The algebra and code are correct and the model is well described in plain and comprehensible language.
In terms of interventions, there are two aims: to reduce the speed of an epidemic (or flatten the curve) and to reduce the size (total number of cases). This manuscript covers the second type of intervention and it would be helpful if this distinction were highlighted.

Line 111 & Equation 8 Given that a dash is being used to denote post-intervention parameters, it might be clearer to use a dot to denote differentials.

It’s a small point, but efficiency is referred to as a proportion in the text but displayed as a percentage. It would be helpful to pick one and stick with it.

Validity of the findings

Line 146 The fit to the data from China as published on 1 April is excellent, but they have just revised the death toll up by almost 50%. The authors state that purpose of these fits is simply to establish the RGM as a reasonable model for the progression of the disease rather than as a predictive tool, so this change doesn’t impact the validity of the findings, but in order to forestall misreporting of the paper it would be helpful to have that made clearer in the discussion of the paper.

Whilst the algebra is correct, it seems counter-intuitive that increasing the intrinsic rate of increase (r) leads to greater efficiency and it would be helpful to see some discussion of what is occurring here.

Line 193 & 199 It would be helpful to have some justification for the parameters chosen for r and alpha in figure 4 because alpha has values that differ by 0.6 and r has values that differ by 0.1. The statement that altering alpha has a greater effect is hard to justify given the disparity between the ranges of values tested. The rationale behind the choices made and for the comparisons really needs to be explained.


Line 218: The authors state that there is a narrow window for interventions to be effective. This is a strong statement and rather implies a cut off time whereas the model supports a statement about effectiveness tailing off. Rapidity is a judgement call and it would be more useful to discuss the timescale of efficiency reduction than a strong but somewhat vague statement.

Figure 4 would be improved by combining all lines onto one graph and representing the mitigation and suppression strategies with a different line type. It appears to represent and interesting point that even right at the beginning of an outbreak when, r is considered to dominate the dynamics, a strategy of suppression appears to have more effect.

Additional comments

This is a timely and helpful manuscript which uses a well understood model to look at how interventions with different effects on the spread of the disease might affect the total number of cases in a pandemic outbreak. There is no suggestion as to what the interventions might be, but the model serves to highlight what outcomes should be sought. The methods are clearly described as are the implications. The authors make some fairly strong claims and these need further justification as described above.

Reviewer 4 ·

Basic reporting

Authors conducted a simulation study on “Modelling fatality curves of COVID-19 and the effectiveness of intervention strategies”. They employed two simple mathematical models and fitted to fatality data in order to investigate the characteristics of fatality curves of COVID-19 and the effectiveness of intervention strategies in five different countries; China, Italy, Spain, Iran, and Brazil. Their study can provide a critical/useful tool for the effectiveness analysis of emerging infectious diseases like COVID-19. However, some critical points from mathematical and epidemiological aspects should be considered for publication in PeerJ.

Experimental design

They used the Richards growth model for China, Italy, Spain, and Iran and the generalized growth model for Brazil (the early stage of epidemics).
Their mathematical models (the Richards growth model (RGM) and generalized growth model) are too simple to capture country-specific COVID-19 transmission dynamics and also country-specific fatalities in the five countries. A mathematical model should be novel enough to explain country-specific confirmed cases and fatality cases. The characteristics of key components in transmission dynamics of COVID-19 should be distinct in each country; for example, population structure (age, ethnic ratios, etc), transmissions (social interactions, clustering etc), interventions (medical infra structures etc), epidemiological features (incubation, infectious periods etc) are very distinct.

Validity of the findings

Authors should further investigate clear relations between infected (confirmed) cases and fatality cases (again, country-specific analysis should be done). The impacts of Interventions such as social distancing, quarantine and intensive treatment (hospitalized) are different on infected cases and fatality cases. This issue also needs to be validated as well. Therefore, it is insufficient for the validity of the epidemic outputs based on the two models the authors suggested. Therefore, the authors should clearly state significant contributions of epidemiological aspects using proper mathematical models.

---

## Round 0.2 · accepted · Accept

Thank you for making the requested changes to your manuscript.

Reviewer 2 ·

Basic reporting

No comment

Experimental design

No comment

Validity of the findings

No comment

Additional comments

The quality of the manuscript has significantly improved and more comprehensive after the revision, especially by including more countrys and longer time in the modeling.

·

Basic reporting

No Comment

Experimental design

No Comment

Validity of the findings

No Comment

Additional comments

The manuscript has bee revised in the light of the reviewers comments and the clarity of the message much improved. The findings remain valid and the links to mechanistic models set the work solidly in context. The simplicity of the model and language make the manuscript accessible to those making policy decisions around new outbreaks of disease.

---

## Author Rebuttal · Round 0.2

**ufpe** UNIVERSIDADE FEDERAL DE PERNAMBUCO          RAYDONAL OSPINA

Departamento de Estatística, CCEN
**CAST** - Computational Agriculture
Statistics Laboratory
Universidade Federal de Pernambuco
Cidade Universitária
Recife/PE, 50740-540
fone/fax: [81] 2126-7437
e-mail: `raydonal@{de.ufpe.br, castlab.org}`

Recife, May 15, 2020

Editor
Dear Editor,

Please find enclosed a revised version of our manuscript ID 47427, which we are resubmitting for publication in PeerJ. We thank all four Reviewers for a careful reading of our manuscript and for providing constructive criticism and helpful suggestions. We have thoroughly revised the manuscript in light of the many useful comments made by the Editor and Reviewers (Round 01). We believe that we have satisfactorily addressed all concerns raised by the Reviewers and have taken up their suggestions whenever appropriate. A detailed, point-by-point response to the Editor and Reviewers is enclosed. We thus hope that the manuscript will now be suitable for publication in PeerJ. Please note that we have included a new author, Arthur A. Brum, who is a PhD student in our group. Mr. Brum has helped us with the new graphs, investigated the connection between SIRD and the Richards models, which informed our new discussion on this topic, and contributed to the overall discussion of the revised manuscript. His contributions thus merit recognition of co-authorship.

Please find enclosed the comments to the reviewers.

**Ref.: Manuscript**
**Title:** Modelling fatality curves of COVID-19 and the effectiveness of intervention strategies
**Research Article:** ID 47427
**Journal:** PeerJ - Life & Environment

**Author(s):** Giovani L. Vasconcelos, Antônio M. S. Macêdo, Raydonal Ospina, Francisco A. G. Almeida, Gerson C. Duarte-Filho and Inês C. L. Souza

Yours sincerely,

Raydonal Ospina
Associate professor

Modelling fatality curves of COVID-19 and the effectiveness of intervention strategies
We thank all the Reviewers and the Editor (Round 01) for a careful reading of the manuscript and providing constructive comments, which helped us to improve the manuscript presentation. We were quite pleased by the Reviewers's generally positive assessment of our work. We were particularly glad that the Reviewers found that our contribution "would be of significance to the community fighting against COVID-19" (Reviewer 2), that "this is a timely and helpful manuscript" (Reviewer 3), and that our "study can provide a critical/useful tool for the effectiveness analysis of emerging infectious diseases like COVID-19" (Reviewer 4). In light of their helpful comments, we have modified the manuscript accordingly to respond to all issues addressed, as explained below in our point-by-point response to the Editor and the Reviewers.

Editor's and Reviewers's comments are shown in **bold,** and our responses appear below the respective comments. We have incorporated in several important ways the Reviewer's suggestions to clarify and improve the paper. We thus hope that they all now agree that the manuscript is suitable for publication in PeerJ.

**COMMENTS**

**Editor comments (Philip Kass)**
**MAJOR REVISIONS**

**Your manuscript elicited a wide range of opinions about its validity, especially about the appropriateness and the simplicity of the models you selected. Although you will need to respond to all the author comments in detail, here are ones that I want to particularly highlight:**

**1. You need to better explain how interventions affect models, and justify your choices of r and alpha. This means tying in the mathematical components of your models with the actual epidemiologic importance of them.**

**Answer:** In the revised version of the manuscript, we have strived to explain and clarify how our approach based on the Richards growth model is related to mechanistic epidemiological models, such as SIR-type models; see, e.g., new discussions in pages 3 and 8. We have also explained in more detail the role of the parameters $r$ and $\alpha$ in affecting the efficiency of interventions and how such effects make epidemiological sense; see page 8.

**2. Add other countries (South Korea, Singapore), and address where model does not do an effective job. One reviewer indicated that your models do not work well for South Korea. If this is true, then this should be explored and explained, as it is important to know under what circumstances (and countries) the models do and do not predict mortality well.**

**Answer:** We have now included not only South Korea but also other countries, such as France and Germany, to better support our approach, and in all cases the agreement between the mathematical model and the empirical data is remarkable. In particular, we point out that that the fatality curve of South Korea is indeed well described by the Richards model, contrary to what anticipated Reviewer 1. We also now explain in more detail the circumstances when the Richards model is not expected to work, see page 6; one such instance is the case of Brazil, which is still in the early growth regime, as shown in Fig. 3 and discussed in the text.

**3. One reviewer made this important comment: "Their mathematical models (the Richards growth model (RGM) and generalized growth model) are too simple to capture country-specific COVID-19 transmission dynamics and also country-specific fatalities in the five countries." Please provide a strong justification for your models in addressing this comment, because interventions will be country-specific, so models should be specific to countries.**

**Answer:** In the revised manuscript we have emphasised that the Richards growth model is a valid epidemiological model, albeit a phenomenological one. It offers a simplified but complementary tool to describe epidemic data, in addition to, say, agent-based and mechanistic (SIR-type) models. To strengthen the connection between the Richards model and mechanistic models, we have now included additional discussion as how the parameters of the Richards model can in principle be related to the epidemiological parameters of SIRD models; see page 3. In this view, the parameters of the Richards model are expected to capture (in an effective and simplified manner) some basic aspects of the underlying epidemic dynamics as well as to reflect the responses adopted by the

respective countries to fight the epidemic. In other words, countries that have different population structures and epidemiological parameters can in principle be described by different sets of RM parameters. The good agreements between the model and epidemic data from the different countries considered in our study are a testament to this fact.

In summary, our view about phenomenological growth models is well summarized by the following quote from Chowell et al. (2016): "Phenomenological models represent promising tools to generate early forecasts of epidemic impact particularly in the context of substantial uncertainty in epidemiological parameters." This is particularly true of the COVID-19 pandemic about which there is and great deal of uncertainties.

**4. Another reviewer wrote: "This is a basic exercise in curve fitting and does not provide any insight to the nature of the epidemic. Since it is a phenomenological model, it is very strange to interpret a change in parameters as interventions. Interventions should modify the mechanisms of disease spread in certain ways, but such mechanisms are not considered here. Even worse, in reality there were a sequence of progressively applied interventions in each of the considered countries." Again, your models should be more than curve-fitting exercises, so it will be important to integrate the epidemiology into the statistical models. Otherwise, your models will be exercises into historic curve fitting, but will not help predict how interventions can affect future mortality in countries.**

**Answer:** As we already explained above, and also emphasise in the revised manuscript, there is much more to phenomenological growth models than merely "curve fitting". For instance, it is possible in principle (working in progress) to establish a sort of map between the parameters of the Richards model and the epidemiological parameters of the SIRD-type models, as we now mention in page 3. It is important to emphasise, however, that the effectiveness of growth models stems precisely from their simplicity, as they often allow for closed form solutions. Because of this (and other properties), they have been successfully applied to epidemic data from other epidemics, such as Zika (Chowell et al., 2016) and influenza (Burger et al., 2019). Veering into a full mechanistic model would hinder one of the main goals of our work, namely to study the effectiveness of intervention-like measures in a more quantitative, mathematical fashion. As for describing interventions, we now explain more clearly that changes in the "mechanisms of disease spread" would naturally lead to changes in the model parameters, so that it makes epidemiological sense (in the context above) to associate "interventions" with possible changes in the model parameters; see, e.g., discussion in page 8.

**Reviewer 1 (Anonymous)**

**Basic reporting**

**Comment:** The paper is well written and presented.

**Answer:** We thank the reviewer for his positive assessment on the presentation of our paper.

**Experimental design**

**Comment:** The authors fit Richard's model to cumulative death data of COVID-19. This is a rather flexible family of curves that can be fitted to practically any sigmoid curve, and cumulative epidemic data often (but not always) have a sigmoid shape. This is a basic exercise in curve fitting and does not provide any insight to the nature of the epidemic.

**Answer:** We agree with the referee that the Richards model allows for a somewhat "flexible family of curves." In fact, the model was devised to do just that, as the standard logistic (or Verhulst's) model is less flexible and often incapable of fitting epidemic data. However, it is not true that the Richards model can fit "practically any sigmoid curve" and by extension most "cumulative epidemic data". As discussed in the manuscript (see also the next point below), there are usually epidemiological reasons as to why the Richards model may fail to fit a particular dataset; for instance, the epidemic may still be in an early stage or the empirical data may display a somewhat sudden change in trend (perhaps owing to a severe intervention or data revision). In other words, even when the model fails, it may nonetheless provide some "insight into the nature of the epidemic" under study. In fact, there is now a considerable agreement on the usefulness of phenomenological growth models as complementary tools to describe epidemic data. This view is well put in the following quote: "Phenomenological models represent promising tools to generate early forecasts of epidemic impact particularly in the context of substantial uncertainty in epidemiological parameters;" which is taken from a new reference, Chowell et al. (2016), that we have included in our discussion of the Richards model in page 3.

**Comment:** Since it is a phenomenological model, it is very strange to interpret a change in parameters as interventions. Interventions should modify the mechanisms of disease spread in certain ways, but such mechanisms are not considered here.

**Answer:** To some extent, all epidemiological models, e.g., agent-based, compartmental, and growth models, are ultimately "phenomenological models," as one does not really have access to the actual epidemic individual mechanism. In all cases, empirical data are needed to calibrate the parameters of the model under consideration. The same is true for growth models based on a single ODE, such as the Richards model and its variants. In this sense, the parameters of the phenomenological growth models are expected to capture (in an effective and simplified manner) some basic aspects of the underlying epidemic dynamics. Changes in the "mechanisms of disease spread" would naturally lead to changes in the model parameters. It is in this sense that we associate "interventions" with possible changes in the model parameters, assuming of course that the growth model is still valid after the interventions. Furthermore, to strengthen the connection between the Richards model and mechanistic models, such as SIR-type models, we have included additional discussion as how the parameters of the Richards model can in principle be related to epidemiological parameters; see page 3.

**Comment:** Even worse, in reality there were a sequence of progressively applied interventions in each of the considered countries.

**Answer:** At the very beginning of our manuscript, we explicitly acknowledge that some countries considered progressive interventions, but eventually were forced to adopt more drastic measures, such as mandatory lockdown. It is reasonable to assume, therefore, that such stringent measures would have a more profound impact on the spread of the disease, to the extent that these interventions could be represented—in a simplified manner—as a change (at a particular time) in the model parameters. In situations where the interventions are adopted early and sustained during most of the epidemic, the epidemic data could in principle be described in terms of a single model (i.e., without change in parameters); see the case of South Korea discussed below.

It is important to emphasise here that the effectiveness of growth models stems precisely from their simplicity, as they often allow for closed form solutions (see, e.g., a related discussion by Chowell et al. (2016) in the context of the Zika epidemic). This was one of the main reasons for using the Richards model for modeling interventions—which was the main objective of our work—, as we were able to derive for the first time (to the best of our knowledge) an explicit formula to quantify the effectiveness of intervention-like measures.

## Validity of the findings

**Comment:** There are no real scientific findings in this article. The Richard's model spectacularrtly fails on Korean data (which was not presented here).

**Answer:** It is not clear to us why the reviewer expected that the Richards model would fail in the case of South Korea. Probably the referee had in mind the Korean data from an earlier stage of the epidemics, when the Richards model was not expected to apply anyway. We now show in the revised manuscript is that the Richards model describes very well the current fatality curve of South Korea; see Fig. 2(f).

**Reviewer 2 (Anonymous)**

## Basic reporting

**Comment:** I find the basic reporting of the study to be very descriptive and realistic. It made me feel that I had a good understanding of what was going to be examined. The language of writing is easy to understand even to the audience outside computational/mathematical modeling field.

**Answer:** We thank the reviewer for his positive general assessment of our paper. In particular, we appreciated that the referee found our "study to be very descriptive and realistic."

## Experimental design

**Comment:** It's interesting to see the modeling the COVID-19 development of several countries in different stages, including China, Italy and Iran, which brings a nice selection of target subjects.

I believe it would be also reasonable if you could include more countries like Singapore and/or South Korea, especially in the "intervention" part of the experiment, as these countries have taken relevant approaches to control the virus situation

**Answer:** We thank the reviewer for the suggestions. We have now included not only South Korea but also other countries, such as France and Germany; and in all cases the agreement between the model and the empirical data is remarkable; see Fig. 2. Singapore, however, has not been included in our study, because it has had so few deaths as of this writing (20 deaths) that the analysis would not be statistically relevant.

## Validity of the findings

**Comment:** I believe that the findings are valid, but the further significance needs to be considered. Is there existing experiments with similar approaches, in dealing with COVID-19 or even non-COVID-19 respiratory infectious disease like influenza? From my point of view, an extra content comparing the current research with other relevant modelings of respiratory infectious disease would be of great help to shed light to the community fighting against the COVID-19.

**Answer:** We again thank the reviewer for the suggestions. We have now included additional discussion on phenomenological growth models as applied to other epidemics. In particular, in page 3 we now state that "phenomenological growth models have been successfully applied to other epidemics, such as Zika and influenza, which makes them good candidates for describing the ongoing COVID-19 epidemic, where there is still substantial uncertainty in the epidemiological parameters." We have also expanded on the connection between the Richards model and compartmental models of the SIRD-type; see page 3.

## Comments for the Author

**Comment:** The current manuscript described the of establishment of a mathematical/computational model of the COVID-19 in aspects of fatality and how it respond to interventional strategies. I believe it's a well written article and it follows nicely to the scope of PeerJ journal publication. The topic is interesting and it could help improve the understanding of the disease development and provide sights to prevention and responding measures. I believe the the submission would be of significance to the community fighting against COVID-19, and how helps draw the attention of disease/disaster response community to the computer science and simulation world.

**Answer:** We appreciate very much the reviewer's overall positive assessment of our work. We were particularly pleased that the referee considers that our contribution "would be of significance to the community fighting against COVID-19" and that it "helps draw the attention of disease/disaster response community to the computer science and simulation world."

**Reviewer 3 (Katharine Preedy)**

## Basic reporting

**Comment:** The paper follows a format close to the Standard structure, there is a background section but no Introduction replaced with a Background section and the materials section replaced with a section data. It might be worth reverting to having an introduction rather than background section to emphasise the point that this paper is about assessing interventions rather than predicting the course of specific outbreaks and that those fits are designed to validate the use of the RGM. In general, however, the structure is logical and clear, the language is approachable and appropriated, there are sufficient references.

**Answer:** We thank the reviewer for the positive assessment of our work and for the helpful suggestions. We have now adopted the standard section structure expected for PeerJ research articles.

**Comment:** The figures are reasonable, though the use of blue and red will make it difficult for those who need to print and view the papers in black and white. Using colours which have different intensities or different line types or widths, or symbols might help with this.
The raw data has been supplied and the code is correct.

**Answer:** We have followed the reviewer's suggestion and used different symbols or line type whenever the issue of legibility might arise in a P&B printed version of the manuscript; see, e.g., figures 1 and 4.

**Comment:** Line 85 – confirmed deaths are more reliable, but "much more reliable" is a strong statement and would need some justification, particularly given that the discussion suggests that timing is critical and there are known to be variable delays in the reporting of deaths

**Answer:** We agree with the referee that the previous wordings ("much more reliable") to refer to confirmed deaths was unjustifiably strong. We now refer to the number of deaths attributed to COVID-19 as "somewhat more reliable"; see page 2. We also have provided additional discussions about the possible uncertainties (under-reporting, delays in reporting, etc) concerning the statistics of deaths from COVID-19; see also page 2. Taking all these factors into consideration, the mortality data can nevertheless be considered as a more reliable statistics, and one that "bears a relation to the dynamics of the number of infections." We thus argue that is quite justifiable to use this type of data in modeling the COVID-19 epidemic.

## Experimental design

**Comment:** **Whilst based on a mathematical model, the focus of the manuscript is on the use of that model to investigate interventions into a pandemic thus it fits well with the scope of the journal. The subject is topical and the research timely. The algebra and code are correct and the model is well described in plain and comprehensible language. In terms of interventions, there are two aims: to reduce the speed of an epidemic (or flatten the curve) and to reduce the size (total number of cases). This manuscript covers the second type of intervention and it would be helpful if this distinction were highlighted.**

**Answer:** We have improved on the discussion of the nature of the interventions modelled by the Richards model (RM). In fact, the subsection about intervention efficiency (in the section of Results and Discussion) has been thoroughly rewritten in light of the many helpful comments from the reviewers and some additional research of our own. For instance, our preliminary results on the connection between SIRD models and the RM indicate that replicating a particular intervention usually might require a combination of changes in both $r$ and $\alpha$. (This is still ongoing research that will be reported in subsequent work, but it helped us to discuss the connection between interventions and the RM parameters in a more informed way.) In this context, we no longer make a specific distinction between "mitigation" and "suppression," which was admittedly a bit premature. Furthermore, we now clarify, as suggested by the referee, that interventions have basically two aims: (i) to "flatten the curve" of daily deaths and (ii) to "bend the curve" of total of deaths; see bottom of page 4. We also point out that the intervention considered in our study are of the latter type.

**Comment:** **Line 111 & Equation 8 Given that a dash is being used to denote post-intervention parameters, it might be clearer to use a dot to denote differentials.**

**Answer:** We thank the referee for pointing this possible source of notational confusion. It has now been fixed, as suggested.

**Comment:** **It's a small point, but efficiency is referred to as a proportion in the text but displayed as a percentage. It would be helpful to pick one and stick with it.**

**Answer:** Again we thank the referee for pointing this possible conflict of terminology. We now use percentage, when referring to intervention efficiency, since the beginning.

## Validity of the findings

**Comment:** Line 146 The fit to the data from China as published on 1 April is excellent, but they have just revised the death toll up by almost 50%. The authors state that purpose of these fits is simply to establish the RGM as a reasonable model for the progression of the disease rather than as a predictive tool, so this change doesn't impact the validity of the findings, but in order to forestall misreporting of the paper it would be helpful to have that made clearer in the discussion of the paper.

**Answer:** For all countries considered in our study, we now use data up to May 8, 2020. This includes the revised mortality data for China, which now has a jump on day 84 since the first death; see Fig. 1. Of course, no model can deal with such an extraneous discontinuity. In order to mitigate this spurious effect we have adopted an ad hoc procedure to make the data amenable to a statistical fit, if only as a test of the model. More specifically we multiplied the data prior to the revision by an appropriate factor, so as to "weld" the two sides of the curve at the jump; see text on page 5. (This is perhaps a worst-case scenario with respect to the true unknown curve.) We then applied the Richards model to this 'renormalised' Chinese data, and the agreement was again very good.

**Comment:** Whilst the algebra is correct, it seems counter-intuitive that increasing the intrinsic rate of increase (r) leads to greater efficiency and it would be helpful to see some discussion of what is occurring here.

**Answer:** We agree with the reviewer that the original exposition about the connection between changes the model parameters and interventions was too condensed. We have now improved on this discussion considerably, as already indicated above. For example, we no longer make a distinction between "mitigation" and "suppression" as being associated with variations of only one of the parameters, respectively. In fact, we point out that it might take a combination of changes in both parameters, $r$ and $\alpha$, to mimic a given real intervention; see page 8. (This is subject of current research.) This does not alter in any way the main conclusions of the manuscript; for instance, that for any intervention the window of opportunity is somewhat narrow; see below. Concerning the role of the parameter $r$, in particular, we now explain, see page 8, that increasing $r$ (with $\alpha$ fixed) implies a smaller inflection time $t_c$, which in turn leads to a smaller $K'$. It is in this sense that a larger $r$ (for $\alpha$ fixed) implies greater efficiency. Similarly: the larger the $\alpha$ (for $r$ fixed), the sooner the curve "bends" toward the plateau, thus yielding a lower final death toll.

**Comment:** Line 193 & 199 It would be helpful to have some justification for the parameters chosen for r and alpha in figure 4 because alpha has values that differ by 0.6 and r has values that differ by 0.1. The statement that altering alpha has a greater effect is hard to justify given the disparity between the ranges of values tested. The rationale behind the choices made and for the comparisons really needs to be explained.

**Answer:** The choice of parameters' values were in fact merely illustrative, since an analytical formula for the intervention efficiency is given and the readers can in principle make their own evaluation. However, the fact that changing $\alpha$ (for $r$ fixed) has a stronger effect in bending the curve than changing $r$ (for $\alpha$ fixed) can be verified by the dependence of $t_c$ on both $r$ and $\alpha$. As a matter of fact, the control in the asymmetry of the "$S$-shaped" curve, and thus the value of $t_c$, was the original motivation for introducing the parameter $\alpha$ in the Richards model, which reduces to the symmetric logistic model for $\alpha$ =1. It is thus natural to expect a stronger effect in changing $\alpha$ (for $r$ fixed) than changing $r$ (for $\alpha$ fixed). We inserted this explanation in the revised text in the penultimate paragraph in page 8.

**Comment:** Line 218: The authors state that there is a narrow window for interventions to be effective. This is a strong statement and rather implies a cut off time whereas the model supports a statement about effectiveness tailing off. Rapidity is a judgement call and it would be more useful to discuss the timescale of efficiency reduction than a strong but somewhat vague statement.

**Answer:** We again agree with the reviewer that our discussion about the 'window' for interventions was too brief. We have now expanded this discussion and given some quantitative figures as to how rapidly the efficiency decreases with the adoption time. For example, we show that, typically, if an intervention is delayed by ten more days, from the time it would give an efficiency of 80%, the efficiency drops to 50% or below; see page 8. This shows that delaying interventions beyond a reasonable early period of time---the so-called window of opportunity--reduces considerably the effectiveness of the intervention. Our analysis also shows, see Fig. 4 and discussion in page 8, that 'stronger' interventions (e.g., with high values of $\alpha$) have wider windows of opportunity, which makes epidemiological sense.

**Comment:** Figure 4 would be improved by combining all lines onto one graph and representing the mitigation and suppression strategies with a different line type. It appears to represent and interesting point that even right at the beginning of an outbreak when, r is considered to dominate the dynamics, a strategy of suppression appears to have more effect.

**Answer:** We thank the referee for this suggestion, which we have taken up. Accordingly, we have combined former Figs. 4a and 4b into a single figure, now labelled Fig. 4a. (We have deleted one of the original curves for readability.) We have proceeded similarly with previous Figs. 5a and 5b, which have now been merged into a single figure, see Fig. 4b. In both cases different interventions have been represented with different line types for legibility, as suggested by the referee.

## Comments for the Author

**Comment: This is a timely and helpful manuscript which uses a well understood model to look at how interventions with different effects on the spread of the disease might affect the total number of cases in a pandemic outbreak. There is no suggestion as to what the interventions might be, but the model serves to highlight what outcomes should be sought. The methods are clearly described as are the implications. The authors make some fairly strong claims and these need further justification as described above.**

**Answer:** We appreciate very much the reviewer's careful reading of our manuscript and useful comments that helped us to clarify and improve the presentation of important aspects of our work. We were also quite pleased with the referee's opinion that the "[T]his is a timely and helpful manuscript."

Reviewer 4 (Anonymous)

## Basic reporting

**Comment: Authors conducted a simulation study on "Modelling fatality curves of COVID-19 and the effectiveness of intervention strategies". They employed two simple mathematical models and fitted to fatality data in order to investigate the characteristics of fatality curves of COVID-19 and the effectiveness of intervention strategies in five different countries; China, Italy, Spain, Iran, and Brazil. Their study can provide a critical/useful tool for the effectiveness analysis of emerging infectious diseases like COVID-19. However, some critical points from mathematical and epidemiological aspects should be considered for publication in PeerJ.**

**Answer:** We thank the reviewer for the generally positive assessment of our manuscript. We were particularly happy with the reviewer's opinion that our "study can provide a critical/useful tool for the effectiveness analysis of emerging infectious diseases like COVID-19."

## Experimental design

**Comment:** They used the Richards growth model for China, Italy, Spain, and Iran and the generalized growth model for Brazil (the early stage of epidemics).

Their mathematical models (the Richards growth model (RGM) and generalized growth model) are too simple to capture country-specific COVID-19 transmission dynamics and also country-specific fatalities in the five countries. A mathematical model should be novel enough to explain country-specific confirmed cases and fatality cases. The characteristics of key components in transmission dynamics of COVID-19 should be distinct in each country; for example, population structure (age, ethnic ratios, etc), transmissions (social interactions, clustering etc), interventions (medical infra structures etc), epidemiological features (incubation, infectious periods etc) are very distinct.

**Answer:** We agree with the referee that growth models, being restricted to a single ODE, cannot accommodate two (or more) variables, such as confirmed cases and deaths. This requires a more general model, such as SIRD-type models, which can also take into account country-specific structures, such as age-structure and economic and social indicators. We note however that there is a close connection between SIRD models and growth models, such as the Richards model (RM); see added discussion in page 3. In the latter model, the specificity of the epidemic propagation in a given population is represented (in an effective manner) by a set of four parameters, namely $r$, $\alpha$, $K$, and $t_c$. In other words, countries that have different population structures and epidemiological parameters can presumably be described by different sets of RM parameters. It is in this sense that the RM can be used to compare epidemic data from different countries, as we have done in the manuscript, as a way of illustrating the general validity of the model. In the name of balance, we have also made clear, whenever necessary, the limitations of the model. We emphasise however that the main goal of the manuscript is not so much to describe particular sets of epidemic data but, perhaps more importantly in the context of the ongoing COVID-19 epidemic, to introduce a quantitative tool to evaluate the effectiveness of interventions.

## Validity of the findings

**Comment:** Authors should further investigate clear relations between infected (confirmed) cases and fatality cases (again, country-specific analysis should be done). The impacts of Interventions such as social distancing, quarantine and intensive treatment (hospitalized) are different on infected cases and fatality cases. This issue also needs to be validated as well. Therefore, it is insufficient for the validity of the epidemic outputs based on the two models the authors suggested. Therefore, the authors should

**clearly state significant contributions of epidemiological aspects using proper mathematical models.**

**Answer:** In light of the reviewer's suggestions, we have expanded the discussion (see page 3) about the connection between the Richards model and SIRD-like models, where other variables, such as the number of confirmed cases, recovered, quarantined, deaths, etc, can be accommodated. However, veering into a full SIRD-type model would hinder one of the main goals of our work, namely to study the effectiveness of intervention-like measures in a more quantitative, mathematical fashion. Different epidemiological models describe different aspects of the epidemic dynamics, at different levels of complexity, ranging from agent-based models to compartmental models to phenomenological growth models. Altogether, these models offer complementary tools that allow one to gain a better understanding of an epidemic, particularly in the case of COVID-19 for which there is considerable uncertainty in the epidemiological parameters. However, as pointed out by the authors of the review article "Mathematical modeling of infectious diseases dynamics" in the Encyclopedia of Infectious Diseases: Modern Methodologies (2007), "a model should only be as complex as needed, depending on the questions of interest." When properly applied (and interpreted) in the epidemiological context, growth models can provide useful insights into the spread of novel infectious diseases. In this sense, they too can be regarded as "proper mathematical models" to describe an epidemic. We hope that, after the revision and expansion of our manuscript, the referee concurs with this view.